# Subfootprint Variability of Sea Surface Salinity Observed during the SPURS-1 and SPURS-2 Field Campaigns

## Frederick M. Bingham

Center for Marine Science, University of North Carolina Wilmington, NC 28403-5928, USA; binghamf@uncw.edu

**Abstract:** Subfootprint variability (SFV), variability within the footprint of a satellite measurement, is a source of error associated with the validation process, especially for a satellite measurement with a large footprint such as those measuring sea surface salinity (SSS). This type of error has not been adequately quantified in the past. In this study, I have examined SFV using in situ ocean data from the SPURS-1 (Salinity Processes in the Upper ocean Regional Studies-1) and SPURS-2 field campaigns in the subtropical North Atlantic and eastern tropical North Pacific respectively. I computed SFV from these data over two one-year periods of intense sampling. The results show that SFV is highly seasonal. I have computed SFV errors in several different forms, a median value of the weekly snapshot error, a total snapshot error, an absolute error of the Aquarius and SMAP (Soil Moisture Active Passive) measurement, a part of that error associated with SFV and a bias due to the skewness of the distribution of SSS. These results are characteristic only of the particular regions studied. However, comparison of the results with high resolution models, and in situ data from moorings gives the possibility of getting global estimates of SFV from these other more common sources of SSS data.

**Keywords:** sea surface salinity; subfootprint variability; precipitation

## 1. Introduction

Satellite measurement of sea surface salinity (SSS) uses a passive microwave radiometer, combined with ancillary measurements of such variables as sea surface temperature (SST), wind speed, atmospheric composition, etc. The retrieval algorithm [1] is a complicated series of steps which takes the initial measurement of brightness temperature and converts it into SSS using the ancillary measurements, models of galactic radiation, an estimate of the complex dielectric constant of seawater [2], etc. The observation is of the top couple of centimeters of the surface ocean. It is difficult to give a precise value of the error in satellite retrieval of SSS, because there are many sources of error, some of which are uncertain themselves. A full discussion is found in [1].

The original idea for the Aquarius satellite, as articulated by [3], was a measurement accurate to 0.2 over a time scale of one month and a spatial scale of 150–200 km. However, thinking about this carefully, this is a difficult standard to test. It is easy enough to average a series of satellite estimates over a month time period, but what to validate that average against is unclear. Validation is the process of comparing satellite-derived geophysical measurements to some independent data. There are generally no estimates of the in situ ocean SSS averaged over a month time period at a 150 km spatial scale. Some of the available Argo synthesis products (e.g. [4]) which are commonly used for this purpose are derived from measurements at an average scale of 1 measurement per 3°×3° square per 10 days. For the most part, validation of satellite SSS has been done comparing values to individual in

situ measurements such as Argo floats [5–7]. Floats come to the surface in relatively random locations, and, if close enough in space and time, can be matched up with satellite samples [8].

Much has been made of the fact that Argo floats generally stop sampling at a depth of about 5 m on their ascent [9,10]. This sampling can miss fresh [11] or, less often, salty [12–14] layers in the top meter or two of the upper ocean.

A largely overlooked source of satellite error is similar to that of representativeness. Satellite measurements that are weighted averages over regions or footprints are being compared to representative values measured at points within those regions. The assumption is that these two sets of measurements are being pulled from the same probability distribution and can be reliably compared. In fact, they are fundamentally different measurements.

Schematically, this is depicted in Figure 1. The satellite estimate of an SSS snapshot at point E is equivalent to averaging the SSS over a region, the central part of which we will call the 'footprint'.

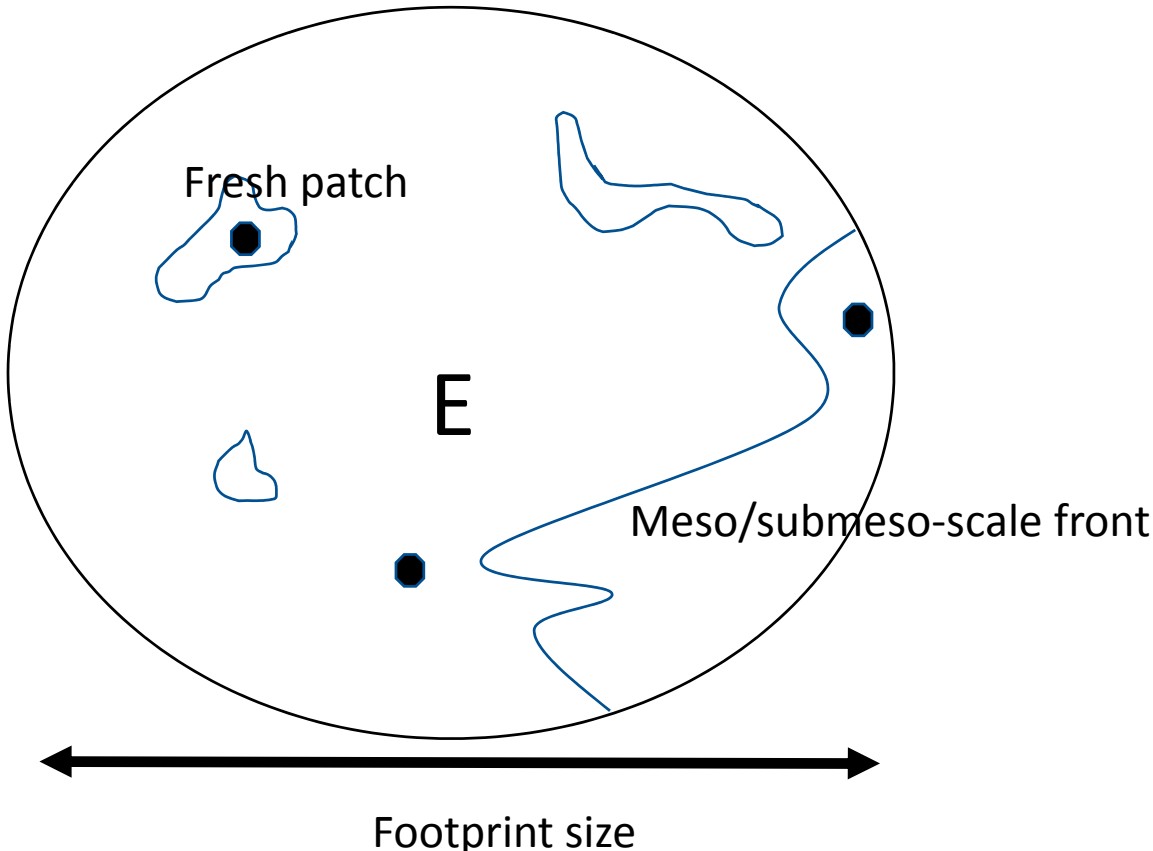

**Figure 1.** Schematic view of a satellite footprint. The large circle is the satellite footprint. 'E' is the point at the center of the footprint where the satellite estimate is made. Dark symbols are potential in situ validation measurements. A mesoscale/submesoscale front snakes through the footprint and a fresh patch, possibly due to recent rainfall, is indicated.

In the case of SSS, the average is weighted, with weighting that decreases in a Gaussian sense with distance from E. Quantitatively, the salinity estimate from the satellite is equivalent to

$$\overline{S} = \frac{\iint wS}{\iint w} \tag{1}$$

where S is the SSS, w is a weight function,

$$w = e^{-\ln(2)*(d/d_0)^2} \tag{2}$$

d is the distance from E to any other point in the region seen by the satellite, $2d_0$ is the so-called footprint size and the integral in (1) is over the entire area seen by the satellite. According to Equations (1) and (2), the footprint is the region from which half of the information that makes up the satellite estimate is taken. Thus, the other half of the satellite estimate is made from information taken from outside the footprint (Figure 2). The footprint for the Aquarius satellite was about 100 km [3]. That for the other two current SSS satellite missions, SMAP (Soil Moisture Active Passive) and SMOS (Soil Moisture and Ocean Salinity), is approximately 40 km [1,15].

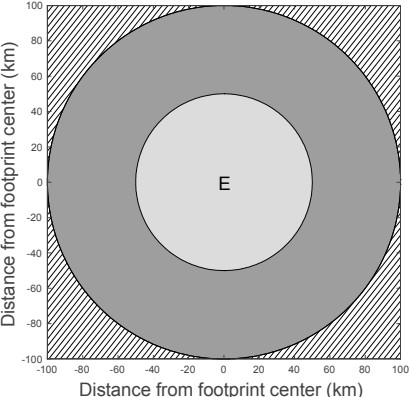

**Figure 2.** Schematic view of the weighting of the satellite measurement for Aquarius whose footprint size is ~100 km [3]. The estimate is made at the central point, 'E'. The footprint (Figure 1; also see locations in Figure 3), is the light gray area from which half of the information to obtain the estimate is taken. The dark gray area is that from which ~44% of the information is taken. Outside the dark gray area, part of which is hatched in this figure, contains about 6% of the information.

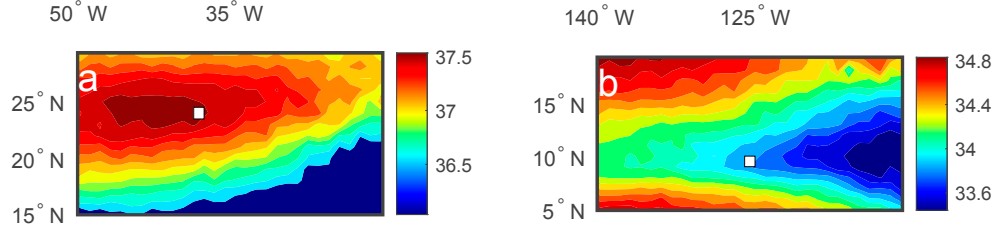

**Figure 3.** Mean SSS over 2011–2014 from Aquarius V5.0 monthly data, for the (**a**) SPURS-1 region in the subtropical North Atlantic, and (**b**) SPURS-2 region in the tropical eastern North Pacific. Color scale is at the right for each panel. Central mooring locations noted with a white box in each panel. The box is 1° on a side, approximately the size of the satellite footprint (Figure 2, light gray).

Within the footprint there may be variability (Figure 1), mesoscale and submesoscale fronts, and fresh or salty patches. All of this variability gets convolved with the weighting function, w, and incorporated into a satellite estimate. In situ measurements like floats (dark symbols in Figure 1) may be present to use for validation, and those may surface into a background field, on one side or another of a front, or into a fresh patch. Floats may also surface in the vicinity of a satellite measurement in space, but not exactly in time. Validation results depend on the details of how the matchups are done.

What is meant by the statement that the two are fundamentally different measurements? The in situ and satellite measurements are generated from different probability distributions. To see how this works, consider the schematic in Figure 1, and suppose there are no mesoscale or submseoscale fronts. That is, the field consists of only a background field, say with value $S_0$, and a set of fresh patches, with varying values $S_f$. [16] shows some examples of synoptic mesoscale SSS fields which match this description reasonably well-though [17] show some different-looking fields. The field is sampled over time by a satellite, whose estimate—given by Equations (1) and (2)—includes all fresh patches in its

view, and by floats, which pop up at random times and locations. Sometimes the floats surface into fresh patches and more often they do not.

Thus, over time, the floats may develop a negatively skewed distribution [18]. That is, they will have a propensity for very low salinity outliers. Assuming the basic statistics of the SSS field are stationary, as the satellite measurements are spatial mean values, the central limit theorem would indicate that they are normally distributed. Thus, the validation process compares normally distributed satellite observations with negatively skewed in situ measurements. The negatively skewed distribution of SSS presents an interesting challenge for validation. Consider a satellite that makes a measurement every few days, and is compared to a float that pops up to the surface in a random location within its footprint. There is some likelihood that the float will pop up into a fresh patch, as the float to the upper left in Figure 1. In that case, the difference between the footprint mean and the float measurement will be large in magnitude and positive. On the other hand, the more likely occurrence is for a float to pop up into the background, not a fresh patch, like the one near the bottom in Figure 1. In that case, the satellite, having made an average over a region that includes fresh patches will make an estimate that is less than the float measurement, and thus the difference between footprint mean and float value will be negative. That is, in the most likely case, the validation process will detect a negative bias of the satellite relative to the in situ measurements due to the fact that the underlying field has a negatively skewed distribution. This bias may differ from location to location because of the changing magnitude of the skewness of the SSS field or the prevalence of fresh patches. All of the above assumes that the SSS field is sampled randomly by the floats. That is, the floats have no sampling bias. A possible complication is that floats may not sample randomly, or the circulation at 1000 m depth, where floats spend most of their time, may not be completely decoupled from the SSS field.

Thus, satellite measurements of SSS may differ from in situ measurements not just because the satellite measurement is noisy, or the retrieval algorithm is imperfect, but because the two measure different quantities. This difference has been called the subfootprint variability (SFV) [19,20], submesoscale salinity variability [21], or representativeness error. This SFV makes up part of the error budget of the satellite SSS measurement, though not included in formal accountings of this budget ([1], their Figure 12). Calling SFV an 'error', in the sense of an incorrect measurement, is a misnomer. It comes from a mismatch of scales of the measurements being compared, and has nothing to do with inaccuracy or bias. Thus, there is a need for better understanding of the size, distribution and variability of SFV.

The most relevant attempt to do this was carried out by [19] and [21], who used in situ thermosalinograph (TSG) measurements to estimate SFV. Volunteer observing ship TSGs measure SSS at a resolution of 2.5 km or better effectively instantaneously, and so are ideally suited to making estimates of SFV. [21] computed standard deviation of SSS (and SST and surface density) at spatial scales of 20 km or less. They found high levels of variability in places one might expect it, most notably western boundary current extensions, and plumes of major rivers such as the Amazon and Congo. [22] found similar results using a global model, but at a larger scale (~100 km).

While these efforts have been groundbreaking, there has been no attempt to examine how SFV might vary over time at a given location, or how SFV might differ as a function of scale. I will examine the first issue, and leave the latter for possible future work. Some idea of how SFV differs as a function of scale was given by [20], who examined it in a high-resolution regional model for the western Pacific and Arabian Sea. They found that about 50% of the variance in SSS is at scales of 50 km or less.

In this paper, I give examples of SFV in two small regions that were heavily sampled to show the time variability of SFV itself, and the probability distributions of SFV on a local scale. I will also examine SFV as determined from a regional high-resolution model, and inferred from a single point moored time series in these same regions. These latter will help us to understand how and whether SFV computed from models and single points can be used to understand the SFV that would be seen by a satellite such as Aquarius. By confining the calculations to Aquarius' ~100km footprint, I focus on that mission alone in this paper, leaving the smaller footprints of the two other current SSS satellite missions

for future work. The two regions, the subtropical North Atlantic and the eastern tropical North Pacific, present very different and contrasting environments for exploring the issues I have discussed.

## 2. Data and Methods

The study regions used here are those where the SPURS-1 and SPURS-2 (Salinity Processes in the Upper ocean Regional Studies) field campaigns were carried out. SPURS-1 took place in the North Atlantic in the subtropical SSS maximum in 2012–2013. It was centered at the location of a central mooring at (24.6°N,38°W) (Figure 3a) [23]. SPURS-2 took place in the eastern tropical North Pacific at the edge of the eastern Pacific fresh pool [24–26] in 2016–2017 (Figure 3b). It was centered at the location of a central mooring at (10°N,125°W) [27]. Each of these regions were intensively sampled at the surface by instruments such as wavegliders, drifters, and thermosalinographs (TSGs) [28,29]. Since each of these instruments sampled at different frequencies, they were subsampled at intervals of approximately 6 hours to get assumed independent measurements.

See Acknowledgements section for DOIs where all data can be accessed.

Salinity values quoted in this paper are from the practical salinity scale, and have no units [30].

### 2.1. SPURS-1

For the SPURS-1 region, I used surface data from drifters [31], TSGs on five different cruises, and wavegliders. The depth of these measurements varies, from ~20 cm on the drifters to ~5 m for the TSG. The cruises occurred in August/September 2012, September/October 2012, March/April 2013 (two cruises), and September/October 2013.

Drifters were released on all the cruises, totaling approximately 113. The processing of the drifter dataset was described in detail by [31]. It included a large amount of manual editing in an ad hoc manner. [31] do indicate that it is possible to continue salinity observations from a drifter for up to a year with minimal drift.

There were three wavegliders in SPURS-1. They were deployed on the September 2012 cruise, recovered, serviced and redeployed in April 2013, and finally recovered in September 2013. The wavegliders had CTD sensors at 60 cm and 6 m depth, but I only used data from the upper sensor. There is unfortunately no documentation on how the data were processed and what if any quality control was applied. This problem is mitigated by the fact that what is being examined in this paper is short-term spatial variability, for which absolute accuracy of measurement is not as important.

The TSG data undergo a rigorous quality control process that is described on the Rolling Deck to Repository website (http://get.rvdata.us/qa_docs/SAMOS).

All three datasets were combined into one year-long record of SSS measurements, totaling about 8800 measured values within 100 km of the central mooring. Estimates of SFV (see Section 2.5) and weighted mean SSS were computed in 7-day blocks. That is, for each estimate of SFV, all values of SSS from all measurement platforms within a 100 km radius of the central mooring and a 7-day period were combined and used in the computation. These 7-day compilations are what I call 'snapshots' to be thought of as level 2 satellite measurements. The assumption here is that the statistics of the SSS field do not vary much during that time. These 7-day periods were used as a compromise. Using longer periods would alias time variability of the SSS field into spatial variability. On the other hand, using a shorter period would make for values of SFV with fewer data points, and therefore less reliability. In SPURS-1, there were about 100 observations per 7-day block, which means a shorter period could have been used without an issue. However, for SPURS-2, as will be seen below, there were fewer than one-third of the observations than for SPURS-1, which makes blocks shorter than 7 days problematic. I wanted to handle the two datasets consistently. Different length blocks were tried for SPURS-1, with little difference in the results.

SFV was computed relative to the location of the central mooring. I also computed monthly anomalies of both the individual measurements and the weekly snapshots. That is, for each month of the SPURS-1 field campaign an unweighted mean value was computed. That mean value was

subtracted from individual measurements and also from weekly averages to examine distributions of anomalies of each.

Most data from SPURS-1 (and SPURS-2) are collected at shallow depths, 25 cm for the wavegliders, ~50 cm for the drifters [31]. The TSG data are collected at 5 m, but while the ship is underway may be effectively sampling at a shallower depth. Some trial computations were done using the deeper waveglider sensor with slightly different results, lower SFV and smaller errors. As satellites effectively sample only the very surface, the computations done in this paper are only with surface values.

The SPURS-1 field campaign included a central mooring, deployed in September 2012, serviced in April 2013 and recovered in September 2013 [32]. The mooring had a CTD sensor at ~50 cm depth, forming a 12-month record of near-surface SSS. The 6-hour subsampled data were grouped into 7-day blocks.

## 2.2. SPURS-2

For the SPURS-2 region, the data used were a little different from SPURS-1. As there were fewer cruises to the SPURS-2 region, essentially only two, no use was made of TSG data. The SPURS-2 region is more dynamic, and thus Lagrangian assets placed there do not remain very long. For that reason, no drifter data were used either. Thus, the SPURS-2 SFV computation uses waveglider data only. The data were subsampled and blocked the same as SPURS-1, making a total of about 2600 individual observations over about 15 months.

The central mooring data in SPURS-2 [33] were similar to SPURS-1. Quality control of the upper sensor, the one used in this paper, was described by [33]. It suffered sensor drift during the deployment due to biofouling, which has not been corrected in the dataset used in this paper. However, the drift was slow, and, as the data here are grouped into short time blocks, and we are interested in variability, not absolute values, it should not affect the results presented.

The SPURS-2 cruise report [34] where the waveglider instruments were recovered indicate that the PIs believe the data to be of good quality. No further documentation of the procedures for handling the waveglider data are available.

## 2.3. ROMS Simulation

In tandem with the SPURS field campaigns, a high-resolution simulation was performed using the Regional Ocean Modeling System (ROMS) specifically configured for the SPURS-1 and SPURS-2 regions. The model is a nested, data-assimilating version of ROMS [35,36]. The resolution of the nest used for this paper is about 3 km. The model assimilates all available data, including altimetry, SST, Argo floats, etc. Most importantly, the model's atmospheric forcing is given by the medium-range Global Forecast System (GFS) at NOAA's National Center for Environmental Prediction (NCEP). The NCEP GFS has an equivalent horizontal resolution of about 18 km. This is adequate to drive the model, but may not be enough to resolve isolated convective systems which produce small-scale rain puddles and sharp horizontal SSS gradients [21,37,38]. More details about the forcing, data assimilation scheme, the rest of the modeling system, as well as detailed comparisons between this model and observations for the SPURS-2 region, are given by [39].

For the SPURS-1 region, the model run went over a one-year period, 1 January 2012–31 December 2012. For SPURS-2, the model run again lasted one year, 1 February 2017–31 January 2018. Model output was made available as daily average values.

## 2.4. Satellite Data

I briefly make use of two monthly-averaged satellite SSS datasets for comparison, Aquarius V5.0 SMI to compare to SPURS-1, and JPL SMAP (Soil Moisture Active Passive) V4.0 to compare to SPURS-2. Monthly data were extracted from these data collections for the time periods and locations of the SPURS field campaigns. Aquarius data are used to formulate the averaged fields displayed in Figure 3.

## 2.5. Computation of SFV

For each 7-day block of in situ measurements, SFV, σ, was computed as a weighted variance,

$$\sigma^2 = \frac{\sum_C w_i \left(S_i - \overline{S}\right)^2}{\sum_C w_i} \tag{3}$$

where $S_i$ is the surface salinity for each measurement within the block, $\overline{S}$ is the weighted average salinity for that block (Equation (4)), and $w_i$ is the weight given to each observation computed from the distance to the central mooring as in Equation (2).

$$\overline{S} = \frac{\sum_C w_i S_i}{\sum_C w_i} \tag{4}$$

The summation, C, in Equations (3) and (4) is over all the observations in a given 7-day block within a distance of twice the footprint radius of the central mooring location. I show results for a footprint size, $2d_0$, of 100 km, both the light and dark gray regions in Figure 2. As discussed above, this 100 km footprint applies only to Aquarius. Use of a smaller footprint to compare with SMOS or SMAP is more challenging as the amount of observations to use in the comparison is smaller, making for less reliable averages. From the time series of σ for each region, I computed the median value, which is labelled $\sigma_{50}$, as a representative number for the SFV.

An example of a 7-day block, Figure 4, shows a large number of very salty values with a couple of much fresher ones, likely associated with rain events.

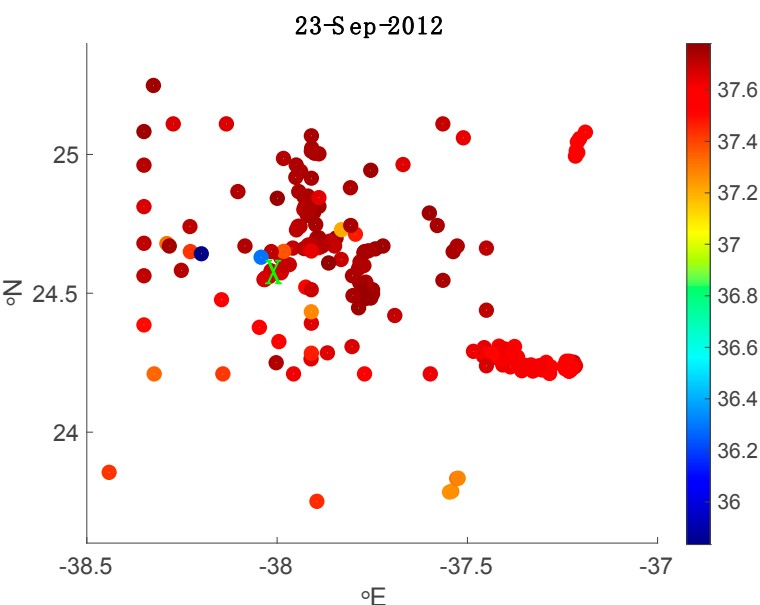

**Figure 4.** Distribution of SSS measurements for the 7-day period surrounding 23 September 2012 in the SPURS-1 region (small circles with color scale at right). The location of the central mooring is indicated by the green letter "X" at (24.6° N,38° W). The value of σ for this date is 0.23.

SFV for the model was computed similarly. However, instead of 7-day values, SFV was computed at each daily time step. To perform the sum in Equation (3), I summed over model grid points within 100 km of the location of the central mooring.

For the central mooring SFV, standard deviation was simply computed for each 7-day block of 6-hour subsampled observations, with no weighting function.

## 3. Results

### 3.1. SPURS-1

SPURS-1 was carried out near the center of the North Atlantic SSS maximum [23] (Figure 3a). Precipitation in the area is small, but has a seasonal cycle, with maximum rainfall during the summer and fall [32,40]. SSS in the SPURS-1 region is shaped by large scale and steady evaporation [40], episodic, seasonal and small-scale rainfall, submesoscale variability [41], and Ekman transport convergence [42]. The time series of SFV from the in situ data (Figure 5a, circles and Figure 5b) shows that it is mostly very small, <0.1, but occasionally increases to much larger values, as high as 0.23 (Figure 4). It also varies in time. It has a seasonal cycle, with higher values in summer and fall than in winter and spring. The histogram of 7-day SFV from the mooring (Figure 5a, thin solid line and Figure 5c), shows that it is much smaller than the in situ values of Figure 5b. (Somewhat larger values were found for increased block sizes, i.e. 14 or 21 days.) The ROMS time series (Figure 5a, red dotted line) are similar to the in situ results, however, the ROMS shows a larger median value of SFV (Figure 5d) than the in situ data, even when sampled at the same times and locations.

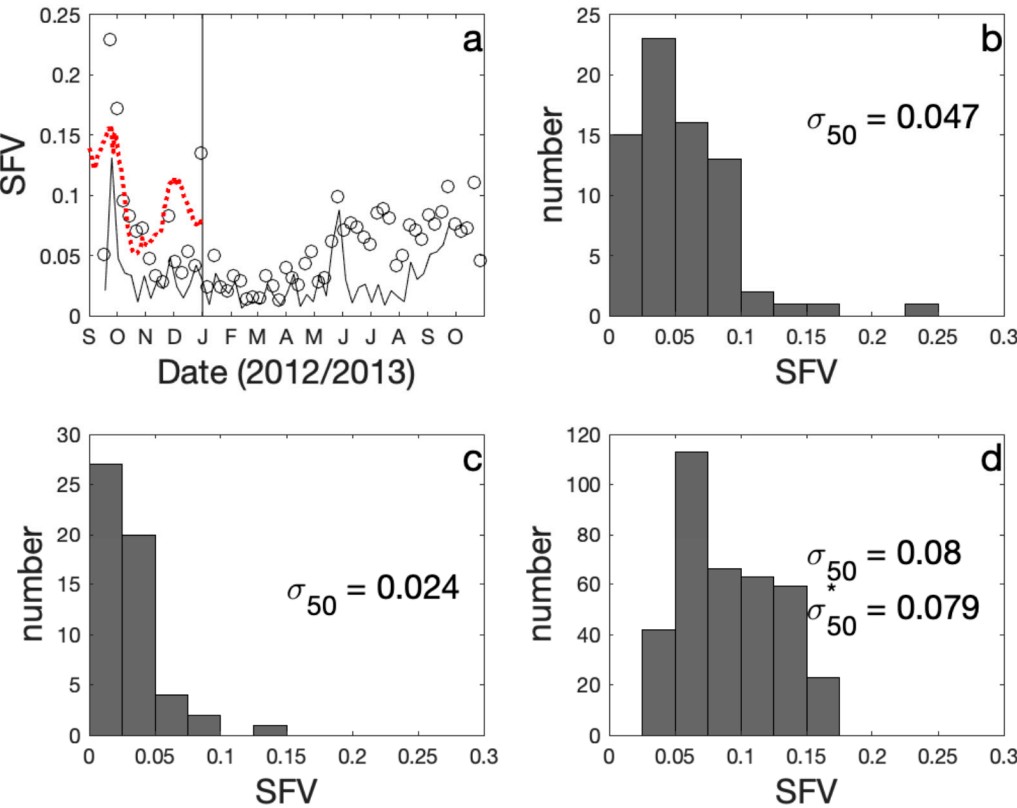

**Figure 5.** (**a**) Circles: Weekly SFV from the SPURS-1 in situ data. Light black line: Weekly standard deviation from the SPURS-1 central mooring. Red dotted line: Daily full SFV from the ROMS simulation. (**b**) Histogram of weekly SFV (circles in panel a). $\sigma_{50}$ is the median value. (**c**) As in panel b, but for the mooring (light black line in panel a). (**d**) As in panel b, but for the full ROMS simulation (red dotted line in panel a). $\sigma_{50}*$ is the median value of the SFV from the ROMS simulation sampled at the same times and locations as the in situ data. Note, less than half of the ROMS simulation SFV time series is displayed in panel a.

To mimic the validation process for the Aquarius and SMAP satellites (e.g. [5–7]), for each 7-day block, I picked out a random in situ measurement, and compared it to the weighted mean computed from all the available measurements. The result, Figure 6, shows points clustered around the one-to-one line with an RMS difference of 0.068. This same computation was repeated 10,000 times, with a different

set of random values and slightly different RMS differences for each repetition. The median RMS difference from all of these repetitions was 0.063. This number could be considered an estimate of the SFV error assigned to validation of satellite SSS at this location for an individual snapshot.

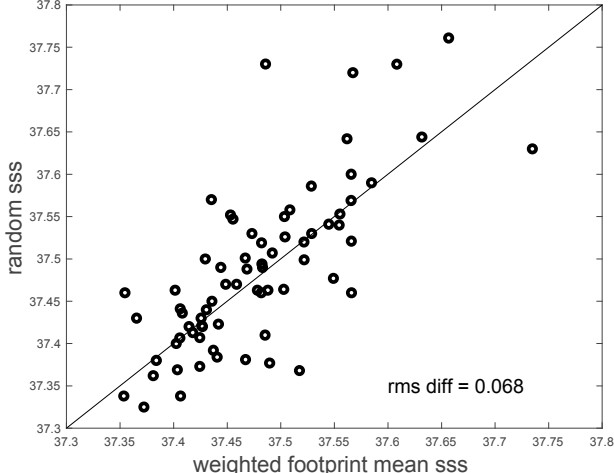

**Figure 6.** Weekly weighted mean SSS for SPURS-1 (*x*-axis) vs. a random value picked out from the ensemble from which the weighted mean was computed (*y*-axis). Each circle is from a different 7-day block during the SPURS-1 period. The line is where the two values are equal.

Monthly-average values of SSS from 10 full months of in situ data were compared with monthly values from the Aquarius dataset at the grid node nearest to the central mooring. The RMS difference was 0.10. Since this comparison is between in situ data and satellite estimates, the satellite estimate incorporates all of the error sources included in the satellite retrieval process as well as the SFV. Thus, the number just quoted is a best guess at the absolute accuracy of the monthly Aquarius values in the SPURS-1 region relative to the intensively measured in situ field. It is well below the threshold set by [3] and incorporated into the original planning of the Aquarius mission, though this result comes from only 10 full months of comparison data. For each month, a standard error for the in situ data (standard deviation divided by the square root of the number of observations) was calculated along with the mean. The median of these 10 standard error values, 0.0024, is an estimate of the difference between satellite and in situ associated with SFV in this region. Note this is much smaller than the total error, about 2% of it.

To illustrate the different natures of the satellite and in situ data, I show the distribution of weekly mean values, and individual measurements (Figure 7). The weekly averages might be something akin to the 'satellite' data as they are averaged over the footprint (panel a), while individual in situ measurements are in panel b). All data are shown relative to their monthly means to remove any seasonal cycle. The 'satellite' data are more or less normally distributed with few outliers, but a slight negative skewness. The in situ data have a negatively skewed distribution, with a number of low outliers with SSS anomaly less than −0.2. The distribution is also strongly peaked at the center of the distribution. This comparison highlights the differing probability distributions from which the two sets of measurements are drawn.

In order to get an idea of the impact of this difference in the SPURS-1 region, the calculation shown in Figure 6 was repeated 10,000 times, each time using a different set of random validation values and computing the mean difference between the random values and footprint mean. This difference itself has a distribution as shown in Figure 8a. There is a slight positive skewness to this distribution, seen in the high outliers. This indicates a tendency for an occasional in situ observation to be made in the middle of a fresh patch, like the blue symbol in Figure 4. The median of the Figure 8a distribution is nearly zero, +0.001. This number could be considered an estimate of the bias due to skewness or

non-normality in the SPURS-1 region, indicating that it has almost no impact despite the apparent skewness of the distribution.

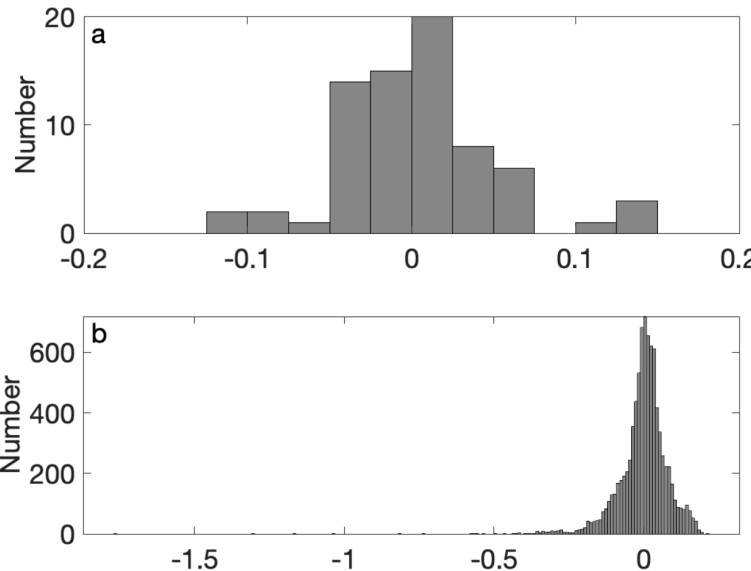

**Figure 7.** (**a**) Histogram of monthly anomaly of weekly weighted mean SSS in the SPURS-1 region. (**b**) Histogram of the monthly anomaly of individual SSS measurements. Note different axis limits in each panel, and the presence of extreme low outliers in panel b.

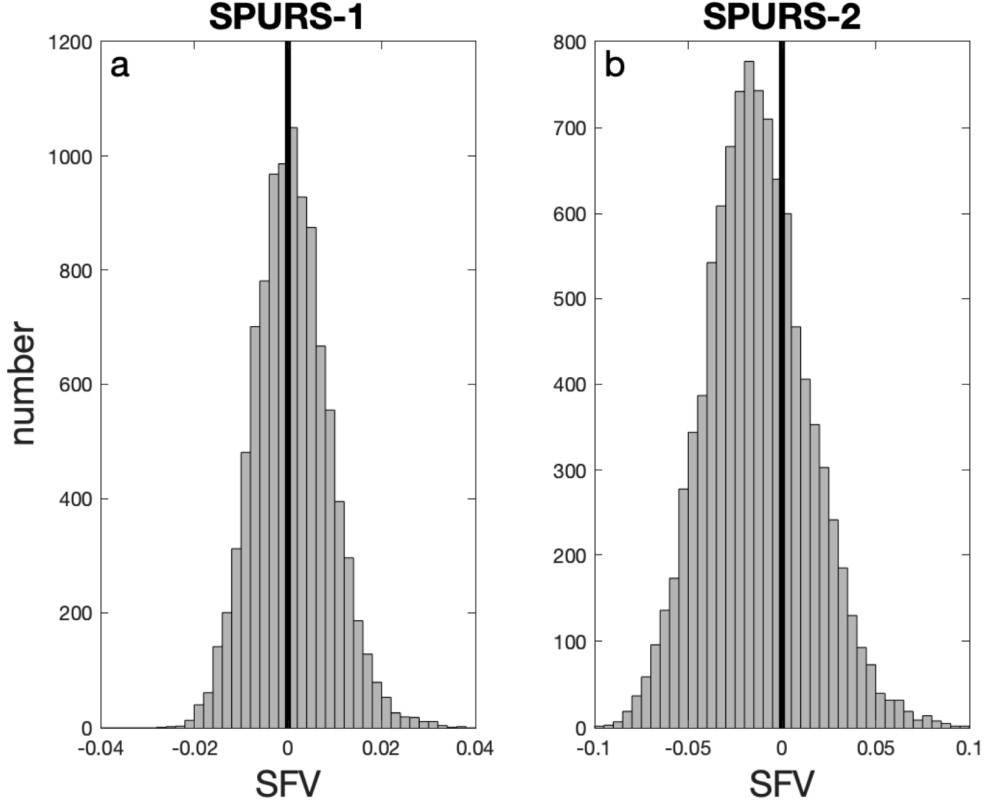

**Figure 8.** Histograms of mean value of difference between footprint mean SSS and random SSS observations. This is the result of the analysis done in Figure 6 repeated 10,000 times, each time with a different random set of in situ observations. (**a**) SPURS-1. (**b**) SPURS-2. Note different axis scales for each plot.

### 3.2. SPURS-2

The same comparisons were done for SPURS-2 as for SPURS-1. SFV again shows a seasonal cycle, with maximum in late summer and fall (Figure 9a). The median SFV from the in situ data (0.13, Figure 9b) matches well with the value from the mooring (Figure 9c). The model (Figure 9d), however, indicates smaller SFV than the in situ data, 0.073 when compared with the in situ data at the same times and places.

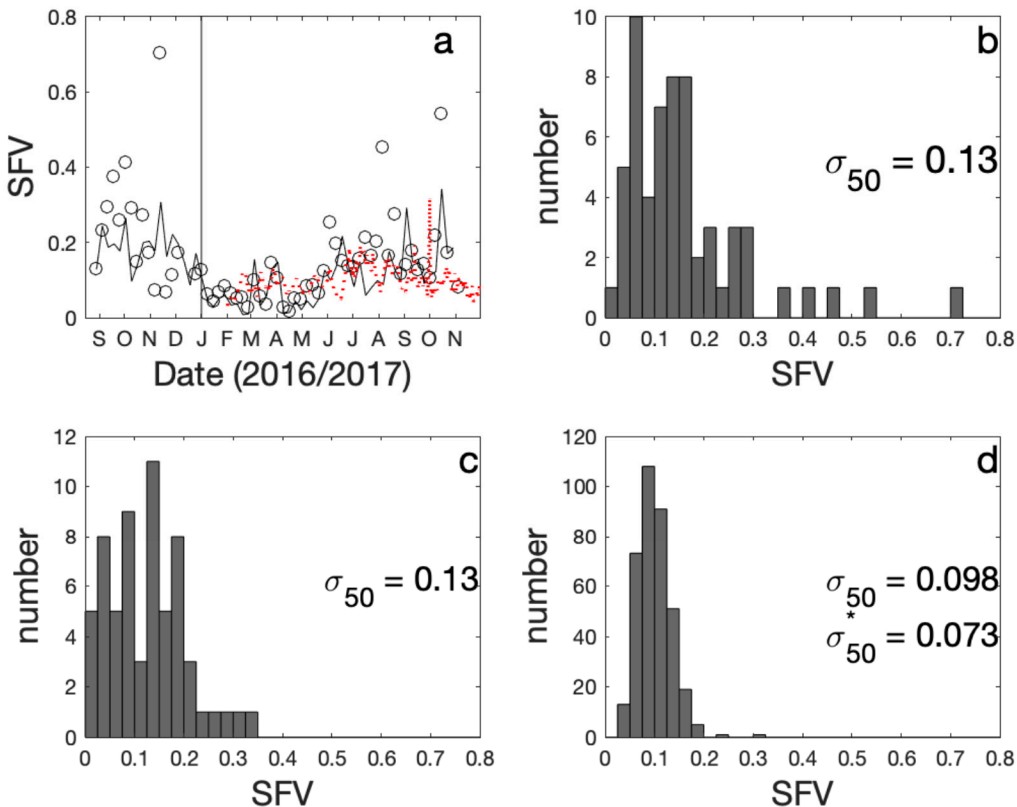

**Figure 9.** As in Figure 5, but for SPURS-2.

The same computation comparing weekly means with random values (i.e., Figure 6) was carried out for SPURS-2, but not shown for brevity. The median RMS difference was 0.196. This is three times larger than the value (0.063) found in the SPURS-1 region, likely reflecting the predominance of rain events, and the increased presence of large-scale fronts [43,44] in the SPURS-2 region vs. SPURS-1. Again, this value, 0.196, is an estimate of the SFV snapshot error in the SPURS-2 region.

As before, monthly average values from the in situ data were compared with satellite values for SPURS-2. This time the satellite data were collected by SMAP. The RMS difference between satellite and in situ was 0.16, a guess at the absolute accuracy of the SMAP monthly values in the SPURS-2 region. In this case, the number comes from 15 months of observations. The median standard error is 0.017, a guess at the error in the monthly values associated with SFV. In the case of the SPURS-2 region relative to SPURS-1, the SFV is a larger portion of the overall error, about 10%. It should be noted that the comparison between Aquarius and SMAP is not quite apt as the footprint size of SMAP is smaller, approximately 40 km.

SSS in the SPURS-2 region is very different from SPURS-1 (Figure 3b). Large scale fronts form and traverse the region seasonally [44]. The presence of the North Equatorial Current and North Equatorial Countercurrent [45] makes advection the largest term in the upper-ocean salinity balance [33]. These currents pull the eastern Pacific fresh pool back and forth in the zonal direction [46]. Rainfall is seasonal, but much more intense [46]. All of these influences work together to make the distribution

of SSS something other than normal (i.e., Gaussian). Examination of the histograms of individual realizations of the SSS field in this region (Figure 10), indicate an amazing variety of distributions. Some increase monotonically in the positive (Figure 10a) or negative (Figure 10b) direction, some are bimodal (Figure 10c), taking their different modalities from either side of a front. Some are more standard in shape, Gaussian, with possible negative outliers (Figure 10d). These are just a few examples. Large negative discrepancies between footprint averages and individual observations tend to occur in the summer and fall—i.e., the rainy season—indicating the influence of low salinity outlier values (Figure 10d for example).

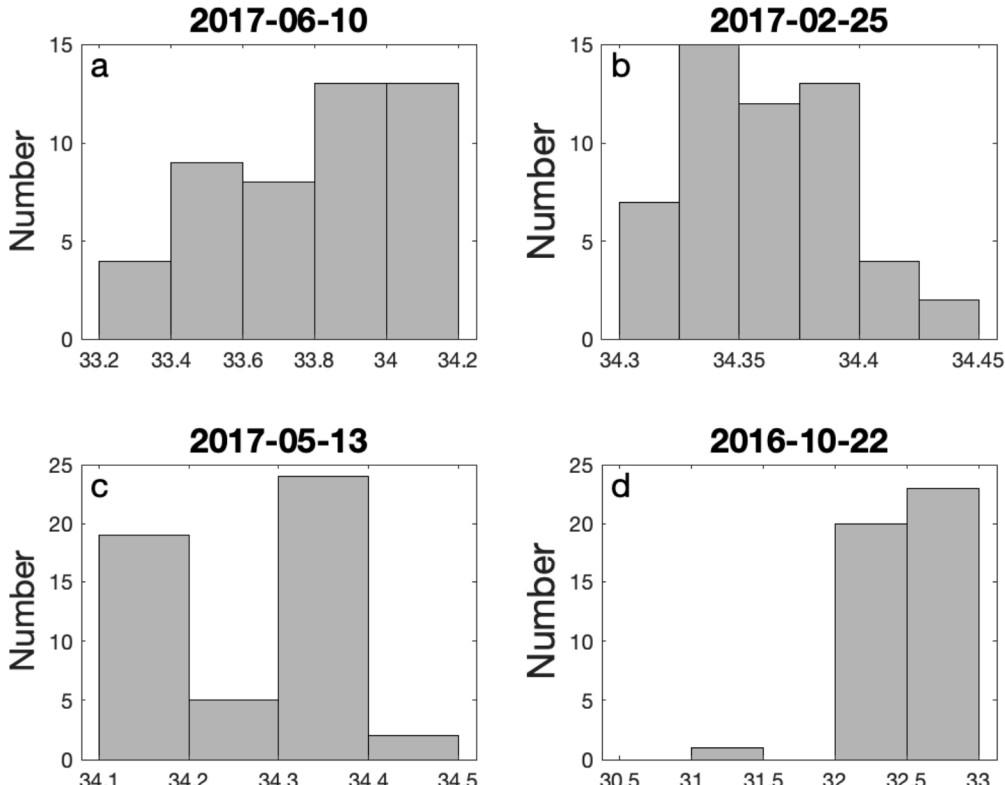

**Figure 10.** Examples of distributions of SSS from SPURS-2 waveglider data. *y*-axes are the number of observations. Distributions are for the 7-day period surrounding the dates shown at the top of each panel, (**a**) 10-June-2017, (**b**) 25-Feb-2017, (**c**) 13-May-2017 and (**d**) 22-Oct-2017. Note inconsistent axis scales.

Despite the variety of distributions as measured synoptically, the full distribution of measurements from the SPURS-2 region is, again, more or less normal for the weekly mean values (Figure 11a), but negatively skewed for individual values, with a number of low outliers (Figure 11b), similar to SPURS-1.

The same set of calculations was carried out for the SPURS-2 data as for SPURS-1 (e.g., Figure 6). Again, this calculation was done 10,000 times to get an idea of how the validation process using randomly-placed measurements would mischaracterize the footprint mean field. The distribution of the difference is shown in Figure 8b. Again, there is a small positive skewness to the distribution indicating a tendency for in situ measurements to be made occasionally within a fresh patch. The big difference between SPURS-1 and SPURS-2 is that this distribution is not peaked at zero, but at a negative value. This number is highly seasonally dependent. It is almost zero in the late winter, spring and early summer, perhaps not coincidentally the times of year when rainfall is minimal. The median is -0.017. This number could be considered an estimate of the influence of skewness on the validation of SSS in this region.

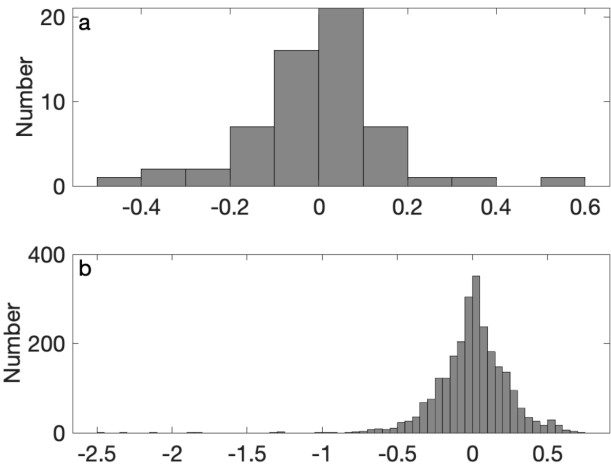

**Figure 11.** As in Figure 7, but for SPURS-2.

## 4. Discussion

Table 1 summarizes calculated median SFV values for the two SPURS campaigns, and the three different datasets. The three are relatively consistent for SPURS-2, but differ substantially for SPURS-1. It is also noted that values from SPURS-2 are much higher than for SPURS-1.

**Table 1.** Summary of $\sigma_{50}$ values from Figure 5; Figure 9

|  | In situ | ROMS | Mooring |
|---|---|---|---|
| SPURS-1 | 0.047 | 0.08 (0.08) | 0.024 |
| SPURS-2 | 0.13 | 0.098 (0.073) | 0.13 |

The very low value from the mooring in SPURS-1 is perplexing as it is inconsistent with the in situ and model values. It appears to have missed a lot of the variability that was measured consistently by the other in situ observations, especially in the summer of 2013. The mooring recorded almost no precipitation during this period ([32], their Figure 4), consistent with the low SFV. By contrast, the seasonal cycle of precipitation in this region shows that precipitation is normally low in spring, but increases substantially in summer and fall [40]. Was the placement of the mooring by happenstance such that any rain events that occurred in the region did not pass near it? This possibility is illustrated in Figure 4, which shows a low outlier value of SSS measured very near, but not right at the mooring location.

The question of using the mooring data time series as a proxy for spatial sampling remains. If time series data at a single point are to be thought of as substitutes for spatial samples within a footprint, the question is how much of the footprint does the mooring actually sample. There are no velocity data publicly available for either central mooring as of the writing of this paper. However, time series of surface-layer velocity for SPURS-2 was published in [33] (their Figure 6 lower panels). The author of this paper has indicated (pers. comm.) that the *y*-axes were mislabeled and should be "m/s" not "cm/s". These data are also 31-day low-pass filtered, though the paper does not state this explicitly. Given typical values from this reference of 0.1 m/s, over 6 hours, the subsampling interval in the present study, water travels 2.2 km. This is smaller than typical decorrelation scales for SSS [47], but large enough to allow for the passage of submesoscale fronts or rain puddles. So, these 6-hour subsamples, I have used in this study may not make for fully independent observations, nor be ideal as a proxy for spatial variability. Nevertheless, they do give some indication of the SFV that would otherwise be unavailable, and may make future quantification of SFV possible using the large amount of mooring data publicly available.

Table 2 summarizes the numerical results that have been presented here, giving estimates of errors associated with SFV for the two regions. The monthly absolute satellite error is within the mission specifications [3], even in a complex region like SPURS-2.

**Table 2.** Summary of error values quoted in the text

|  | SPURS-1 | SPURS-2 |
|---|---|---|
| Snapshot SFV error | 0.063 | 0.196 |
| Monthly absolute satellite error | 0.10 (Aquarius) | 0.16 (SMAP) |
| Monthly satellite error due to SFV | 0.002 | 0.017 |
| Bias due to SSS distribution | +0.001 | −0.017 |

Kao et al. [5] found that the globally-averaged RMS difference between weekly satellite maps and collocated in situ data do not exceed 0.2. It was found that the RMS difference between footprint mean and random observations, what I call the snapshot error, is 0.063 for SPURS-1 and 0.196 for SPURS-2. Their map, Figure 23d in reference [5], shows similar values for the SPURS regions. My calculations and those of [5] are somewhat different in that mine do not include any error associated with the satellite retrieval algorithm. It is clear that SFV-associated errors are functions of space and time, and, in the open ocean away from large-scale fronts such as the Gulf Stream, tend to be largest in the presence of rainfall.

Similar calculations were done with a high-resolution model. In the SPURS-1 region, the model overestimated the SFV (0.08 vs. 0.047), whereas in the SPURS-2 region it underestimated (0.098 vs. 0.13). Perhaps this is due to the fact that, in both cases, the model was driven by a relatively coarse NCEP GFS precipitation field that may not be quite appropriate in either place. Another possibility is that the model does not correctly estimate the SFV that is internal to the ocean, overestimating it in the SPURS-1 region and underestimating in SPURS-2. Also possible is the lack of overlap in timing, especially in SPURS-1 where the model and field campaign only overlapped by about 4 months. This brings up the question of interannual variability of SFV, which is not addressable with the SPURS datasets. It awaits some other more definitive investigation in the SPURS and other regions.

The contrast between the SPURS-1 and -2 regions in terms of the difference between individual observations and footprint averages (Figure 8) is notable. As stated above, the distributions in both regions are somewhat positively skewed, meaning that there is a tendency for occasional in situ observations to be made within isolated fresh patches as in the blue symbol in Figure 4. On the other hand, the overall distribution is peaked near zero for SPURS-1, but at a negative value for SPURS-2. This indicates a general tendency in the SPURS-2 region for the footprint average to be less than individual observations. Why might this be, and why in the SPURS-2 region and not SPURS-1? If there are a lot of small, very fresh patches in the SSS field, the footprint mean estimate would include all of them, whereas the individual observations might or, more likely, might not sample them. Thus, the more small fresh patches there are, and the more intense and isolated, the greater the tendency for the footprint mean to be less than an individual measurement. So, the fact that the distribution of Figure 8 peaks at a value less than zero may be an indicator of the prevalence of isolated fresh patches in the SSS, which is more likely to be the case in the rainfall-dominated SPURS-2 region than SPURS-1.

## 5. Conclusions

In this paper, I have examined SFV using the SPURS-1 and SPURS-2 data collections. SFV was shown to vary seasonally in both locations as shown in Figures 5 and 9. The principal conclusions of this paper are detailed in Tables 1 and 2, which give estimates of errors due to SFV in these two regions. Table 1 gives median values of SFV, 0.047 and 0.08 for SPURS-1 and -2 respectively. Table 2 gives error values in snapshots (0.063 and 0.196), errors relative to monthly satellite-based SSS values (0.10 and 0.16), and errors due to skewness in the distribution of SSS (+0.001 and −0.017). It should be emphasized once more here that the word 'error' may be misleading in this context. The errors result not from measurement inaccuracies or difficulties in retrieval, but from a mismatch in scale between the footprint-averaged satellite and the single-point in situ measurements.

This study brings up many issues surrounding the problem of SFV that remain to be explored. Most important is to determine what causes SFV. More generally, what determines spatial variability of SSS at scales smaller than the footprint of the satellite, which can be 100 km or smaller? Is it variability internal to the surface ocean, or imposed on it by freshwater and momentum flux from the atmosphere? Does the cause of variability depend on spatial scale? From comparison of the two regions examined in this study, rainfall is probably the determining factor because of its constant imposition of small-scale fresh patches [38,39]. However, this hypothesis has not been explored here or elsewhere in any depth. Are rainy regions any more prone to SFV than less rainy ones?

SPURS central mooring data have been used in this study to make estimates of SFV by substituting time variability for spatial variability. While the results were less than ideal for the SPURS-1 mooring, the concept merits further study and use. There have been many moorings placed in the global ocean that have collected the type of data (rainfall and SSS) to determine SFV and its relationship to rain [48]. These mooring data have great potential for future study.

Also of interest is how SFV may be determined by the use of high resolution models, an area of current active research [20]. At present the highest resolution freshwater forcing fields are 18 km. Rainfall often has scales that are much smaller than this [38,49]. If rainfall is the primary determinant of SFV, can it be adequately quantified from a model, even a very high resolution one, with such forcing? More fundamentally, rainfall adds SSS variance to the ocean at scales inherent to the atmospheric forcing. What impact does this addition of variance have on upper ocean circulation and mixing and how might this feedback into the marine atmosphere? How do model results depend on the scales of the freshwater forcing? Such questions are left to be considered by future researchers using the model, in situ, and satellite datasets that have become available over the past few years.

**Funding:** This research was funded by NASA, grant number 80NSSC18K1322.

**Acknowledgments:** I gratefully acknowledge the PIs responsible for collecting the high quality in situ data used in this study, especially L.Centurioni and V.Hormann (SVP-S drifters), J.T.Farrar (mooring) and B.Hodges and D.Fratantoni (wavegliders). I also thank Z.Li for providing me with ROMS output and O.Chkrebtii for many useful discussions.

**Conflicts of Interest:** The author declares no conflict of interest.

**Data Sources:**

SPURS-1

- Drifters: doi:10.5067/SPUR1-DRIFT
- TSGs: doi:10.5067/SPUR1-TSG00
- Wavegliders: doi:10.5067/SPUR1-GLID3
- Central mooring: doi:10.5067/SPUR1-MOOR1

SPURS-2

- Wavegliders: doi:10.5067/SPUR2-GLID3
- Central mooring: doi:10.5067/SPUR2-MOOR1

Satellite

- Aquarius: doi:10.5067/AQR50-3SMCS
- SMAP: doi:10.5067/SMP3A-3SMCS

ROMS

- Available from the author on request

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
