# Peer review of "Subfootprint Variability of Sea Surface Salinity Observed during the SPURS-1 and SPURS-2 Field Campaigns"

_remotesensing, doi:10.3390/rs11222689_

Round 1
Reviewer 1 Report
Sub-footprint variability of sea surface salinity observed during the SPURS-1 and SPURS-2 campaigns by Frederick M. Bingham
This work investigates how oceanic spatial variability in scales less than a satellite footprint impacts the evaluation of the satellite measurement performance. The satellite in mind seems to be Aquarius, that worked from 2011 to 2015, although the methodology could have been applied for SMAP or SMOS characteristics. For investigating sub-footprint variability, the author uses observations collected during two fieldwork campaigns, one in the North Atlantic Ocean and the other in the Tropical Pacific Ocean, which are part of the SPURS project sponsored by NASA.
The MS deals with an essential issue of the remote sensing of sea surface salinity since the satellite missions up to the present date have moderate to low spatial resolutions (40-150 km), while in situ measurements are punctual and captures variability in scales lesser than that. The mismatch between satellite and in situ observations is aggravated because the two systems do not observe the same levels in the water column. Moreover, the knowledge of spatial salinity variability is also reduced because of the historical scarcity of in situ salinity observations on a global scale. Therefore, studies such as present ones should be a good incentive, and quite welcome. I believe the scope of the paper is very appropriate to the Remote Sensing journal.
My comments below aim to improve the readability and presentation, in particular for researchers outside the salinity remote sensing community. While there are many comments, I believe they are minor corrections, and the author will able to fix them quickly, as listed below.
L#26. “SSS (via global model).” My experience shows that this statement may confuse researchers outside the remote sensing community. Many people because similar statements in other MS started to believe that satellite SSS reflects the numerical model ones. I think the remote sensing satellite community must be careful about that and explain better how the modeled SSS enters in the satellite processing.
L#51-52: “that of the change-of-support problem in geostatistics.” Please, explain in the text what that means. I believe many oceanographers do not know anything about it. However, the statement could be suppressed since it is tangential for the present study.
L#57-66: The explanation may confuse the reader because it is not how the satellite ‘measures’ salinity. I think you could say that “it is equivalent to” such that: “The satellite estimate of an SSS snapshot at point E is equivalent to averaging the salinity measurements over a region, which we will call the footprint.” “Quantitatively, the salinity estimate from the satellite is equivalent to:”
Equations 1 and 2: remove the borderline in the left and the top of the equation. They distract the reader.
Figure 2: I know that the author used this Figure in a previous paper, but it can cause some confusion. I think it would be clearer if the white area outside the dark blue circle was also shaded because, according to the caption, that area (outside the dark blue circle) accounts for 6% of the information received. Because it is ‘empty,’ it seems that the information only comes from within the blue/cyan circled areas.
L#102: instead of “very low outliers” should be “low salinity outliers” if I understood correctly.
L#104-117: A missed point here is that the in-situ observations from floats, for example, are not homogeneously distributed in space given the floats are Lagrangian measurements that drift with the circulation. Thus, the in-situ measurements have also their intrinsic sampling biases that will affect the comparison with the satellite.
L#118-119: I did not get the point here. The paragraph discusses the mismatching of satellite and in situ measurements, and this statement seems displaced in relation to the paragraph content. Why would we assume that the SSS is homogeneous and with no large-scale gradients within the footprint?
L#121-123: Writing needs improvement because all of these facts happen at the same time in reality: The satellite algorithms are imperfect; the satellite data is noisy; the in-situ observations are very different from the satellite measurements.
L#125: “error” is not the adequate word because many of the problems, as the author points out, are not due to errors (right/wrong measurements) but due to technological limitations of both in situ and satellite observations since we cannot measure the same thing yet. Both measurements can be accurate and right and even so very different. Therefore, the difference does not fit in a classical definition of error.
L#144-151: Needs to clarify that the SFV that this paper focusing it is for the Aquarius spatial scale (<100 km). This point is not clear, and this makes the readers getting confused in some MS passages. It may be pointed out that the methodology could be applied to SMAP-SMOS scales of 40 km or less. I am unsure why the author did not use for this scale, but maybe there is not enough in situ data to evaluate these scales, or maybe this will be done in another study. Please explain in the text.
L#166-169: The maps show the average SSS between 2011-2016 from the SMOS satellite, but the 1-degree footprint size is from Aquarius, although this information is missing in the caption. Please improve the caption title. Why use SMOS? Please explain.
L#187-197: It would read much better if the SFV were defined here or if this part was moved to subsection 2.5, avoiding the readers needing to jump back/forward to understand what the author is referring to.
A point that is also unclear and should be fixed is why to compute the SFV over seven days since the satellite footprint is almost instantaneous. It takes minutes to the satellite to sweep a large area. I guess the author is thinking in the SFV for the case of comparison between in situ and level-3 Aquarius products. Another possibility is that maybe there is no enough in-situ data in 1-day or less, and the author is assuming that there is no temporal variability over a time window. Please give some explanation in the text.
L#216-228: This section is contradictory because it says that numerical modeling experiments accompany the SPURS campaigns, but for the SPURS-2, the modeling period is posterior to the campaign that ended in 2017. I am unsure if there is some typing mistake. Are the modeling experiment dates, right? If yes, please improve the text.
L#232-236: It should say that the Aquarius for SPURS-1, SMAP for SPURS-2, and SMOS for both.
L#241-254: An explanation of why the SFV is computed over many days is required here. If the idea was simulating as close as possible the satellite view, the period should be of the order of minutes. It also needs to explain why 7-days specifically. I believe this related to the Aquarius repeat cycle. Moreover, the 100 km is the footprint of Aquarius, and that is unclear in the text. You could have used 40 km to simulate SMAP that is still operational.
Figure 4: The figure and caption need improvement. The label should be longitude and latitude, so the reader gets the figure immediately and the letter ‘C’ is not identifiable. Which instruments made the ‘+’ observations?
L#279: I think it is referring to the wrong figure here. Figure 4 does not show SVF. I believe it should be Figure 5a. Check.
Figure 5: It needs improvement. First, Fig 5a should use colors to make the curves visible since even the caption recognizes that the dotted line is barely visible. Also, the heavy dashed line doesn’t look like a line and got mixed with the circles. There is no y-axis’ title in 5a. In 5b-5d: Number of what? The x-axis title ‘pss’ sounds strange. If I understood correctly, the x-axis shows the SFV.
L#296-302: This ‘mimic’ the validation process for level-3 products, but with level-2 (swath) data, in general, a window of +- 12h is used and not 7-days. Please let clear which kind of validation the present work is simulating. Moreover, for SMAP, a radius of 40 km would be the most adequate. What would happen with the SFV values, whether a 40 km radius was used? I don’t think the word snapshot is appropriated in this statement “for an individual snapshot,” given you are computing SFV over a 7-day period.
Figure 6 caption: Improve caption text. There are many random/weighted SSS mean value pairs in the plot, and the description seems to be referent to one dot only.
L#308-315: I suggest changing the word ‘error’ to 'differences' since ‘error’ is not appropriate in the case of comparison between satellite and in situ measurements. Please, also define what is “the standard error for the in-situ data.”
L#317: What ‘it’ refers to?
L#320: It should specify “Level-3” and not satellite-only because the original satellite data are not weekly averages, and this fact may confuse the readers.
Figures 7: Axis titles should be improved. The number of what? Labels for the x-axis are required to make the plot straightforward.
Figure 8: The number of what? Labels for the x-axis are unclear.
Figure 9: Needs the same improvements as Figure 5 (see my comments above).
L#354-366: Replace ‘error’ with an appropriate word (see my previous comments). Define what the “absolute accuracy” is.
L#380: Define what is a “low outlier.” Would it mean low-salinity value outliers? Please clarify the text.
Figure 10: Keep consistency with the other figures — the title is missing in all axis in this Figure.
L#409: Why perplexing?
L#409-417: I believe that ROMS could have been fully explored to better understanding the SFV. How well ROMS reproduces the in-situ observations? Have you sampled ROMS at the mooring position? Would the ROMS simulated mooring time series present the same low SFV?
L#419-429: I did not understand the argument the author tried to present. Please improve this part. Why would a punctual mooring be a proxy for spatial sampling?
L#432-onwards: The ‘error’ terminology throughout the text (e.g., “absolute satellite error”) seems to be in contradiction with the core of the work itself. We cannot talk about errors when we are comparing so many different kinds of measurements. It would equivalent to say that the orange failed because it has a different flavor of an apple. I suggest avoiding the use of the word ‘error’ in this context.
L#473-486: This is not a well-posed question. The SFV, as defined in the present work, is dependent on the satellite footprint. SMAP has much higher spatial resolution than Aquarius, so we can imagine the processes regulating the SFV in Aquarius are different from the ones in SMAP. If, in the future, we could have a 1 km resolution salinity satellite (why not dream?) such as for temperature, and our SFV would have a completely different nature. Therefore, I believe that the word ‘SFV’ is not adequate in the context. Maybe just say what controls the spatial variability of salinity at meso-to-submesoscale. It would fit better to discussion present here, which is not relative to a specific satellite ‘sub footprint’.
L#488-495: I suggest deleting this conclusion section or merging it with the discussion to improve the readability.
Reviewer 2 Report
This is a review of “Subfootprint variability of sea surface salinity observed during the SPURS-1 and SPURS-2 field campaigns” by Frederick M. Bingham. This study addresses an important question when looking at satellite SSS measurements, and that is what is going on within the satellite's subfootprint? The author utilizes many in situ SSS measurements from two different regions that were recently heavily observed to estimate the subfootprint variability (SFV) of SSS. The major conclusions drawn from this study is that SFV is highly seasonal (in both regions) and that it can be a significant source (contributing upwards of ~10% in the SPURS-2 region) of error in the satellite validation process. Overall, the manuscript is well written and scientifically sound. I believe the manuscript is suitable for publication in Remote Sensing after a few minor comments/edits are addressed:
General Comments:
1.) The study utilizes mostly in situ data, but there is very little discussion about what kind of quality assurance/control the in situ data had undergone prior to being used in this study. I realize that there are citations to the data sets, but it would be helpful if a brief description was provided in the text. For example, did any of the instruments suffer from salinity drift?
2.) Most of the in situ calculations for SVF rely on 7-day blocks. A lot of salinity changes can take place within a 7-day time frame, particularly in the SPURS-2 region. Was there any attempt to reduce this time period to perhaps a few days or just one day (like with the model)? I assume the lack of observations as you reduce the time period is a limiting factor, but it would be helpful if a brief explanation could be included discussing the possible strengths/weaknesses of the 7-day time period.
3.) Was there any attempt to include Argo floats or was it only restricted to instrumentation deployed during the SPURS campaigns? I realize Argo's shallowest measurements are a few meters deep and there likely won't be many observations near the central mooring, but it might add a few more measurements to the study, particularly in the SPURS-2 region where only wavegliders were utilized.
Minor Line by Line Comments:
Line 14: Suggest including “respectively” - “… eastern tropical Pacific, respectively.”
Figure 1, Page 2: Suggest adding more to the caption. It is explained in the text, but it would be helpful if the Figure caption included a description of the dark circles in Figure 1 as well as the letter “E”.
Equation 1, Page 2: If the satellite SSS is independent of in situ SSS, then why is in situ SSS included in the salinity estimate from satellite (Equation 1)?
Equation 2, Page 2/3: What is “d” the distance to? From E to the in situ SSS measurement?
Reviewer 3 Report
The subfootprint variability of SSS is important for the validation process of space-borne SSS and is discussed in this manuscript based on SPURS campaigns, moorings and ROMS data. The results are interesting and can contribute to the estimate of the error of space-borne SSS. The manuscript is acceptable after minor revisions.
Specific comments are listed below:
1. The differences of SSS due to variations at different depth also contribute to the subfootprint variability of SSS. Please discuss the proportion of variability of SSS due to variations of depth to that over the surface.
2. The conclusion part is too short. Pleas make this part longer or merge section 4 and section 5.
